# Biomedical Trends in Stimuli-Responsive Hydrogels with Emphasis on Chitosan-Based Formulations

**DOI:** 10.3390/gels10050295

**Published:** 2024-04-25

**Authors:** Weronika Kruczkowska, Julia Gałęziewska, Katarzyna Grabowska, Gabriela Liese, Paulina Buczek, Karol Kamil Kłosiński, Mateusz Kciuk, Zbigniew Pasieka, Żaneta Kałuzińska-Kołat, Damian Kołat

**Affiliations:** 1Department of Biomedicine and Experimental Surgery, Faculty of Medicine, Medical University of Lodz, Narutowicza 60, 90-136 Lodz, Poland; weronika.kruczkowska@stud.umed.lodz.pl (W.K.); julia.galeziewska@stud.umed.lodz.pl (J.G.); katarzyna.grabowska1@stud.umed.lodz.pl (K.G.); gabriela.liese@stud.umed.lodz.pl (G.L.); paulina.buczek@stud.umed.lodz.pl (P.B.); karol.klosinski@umed.lodz.pl (K.K.K.); zbigniew.pasieka@umed.lodz.pl (Z.P.); zaneta.kaluzinska@umed.lodz.pl (Ż.K.-K.); 2Department of Molecular Biotechnology and Genetics, University of Lodz, Banacha 12/16, 90-237 Lodz, Poland; mateusz.kciuk@biol.uni.lodz.pl; 3Department of Functional Genomics, Faculty of Medicine, Medical University of Lodz, Zeligowskiego 7/9, 90-752 Lodz, Poland

**Keywords:** chitosan, formulations, stimuli-responsive hydrogels, smart hydrogels, biomedicine, trends, wound healing, tissue engineering, drug delivery

## Abstract

Biomedicine is constantly evolving to ensure a significant and positive impact on healthcare, which has resulted in innovative and distinct requisites such as hydrogels. Chitosan-based formulations stand out for their versatile utilization in drug encapsulation, transport, and controlled release, which is complemented by their biocompatibility, biodegradability, and non-immunogenic nature. Stimuli-responsive hydrogels, also known as smart hydrogels, have strictly regulated release patterns since they respond and adapt based on various external stimuli. Moreover, they can imitate the intrinsic tissues’ mechanical, biological, and physicochemical properties. These characteristics allow stimuli-responsive hydrogels to provide cutting-edge, effective, and safe treatment. Constant progress in the field necessitates an up-to-date summary of current trends and breakthroughs in the biomedical application of stimuli-responsive chitosan-based hydrogels, which was the aim of this review. General data about hydrogels sensitive to ions, pH, redox potential, light, electric field, temperature, and magnetic field are recapitulated. Additionally, formulations responsive to multiple stimuli are mentioned. Focusing on chitosan-based smart hydrogels, their multifaceted utilization was thoroughly described. The vast application spectrum encompasses neurological disorders, tumors, wound healing, and dermal infections. Available data on smart chitosan hydrogels strongly support the idea that current approaches and developing novel solutions are worth improving. The present paper constitutes a valuable resource for researchers and practitioners in the currently evolving field.

## 1. Introduction

The field of biomedicine is constantly evolving to provide substantial advancements in healthcare. Tremendous progress was made regarding novel and unique strategies, such as advanced materials that are devised, tested, and thoroughly examined to improve safety and effectiveness in various biomedical applications. These materials are carefully engineered with specific physical, chemical, or biological properties to prevent and treat various diseases [1,2]. Hydrogels represent a cutting-edge and highly promising subset of advanced materials that is worthy of in-depth exploration [3].

Hydrogels are three-dimensional (3D) polymer networks with the characteristics of both solids and liquids, allowing for straightforward measurement of their mechanical properties. Since they resemble liquids, processes such as diffusion, absorption, and release of molecules are facilitated [4,5,6]. Hydrogels can be categorized based on various factors, including their source, preparation method, ionic charge, responsiveness, cross-linking, and physical properties [7]. Incorporating natural polymers as their primary components provides benefits such as biocompatibility and biodegradability [6,7,8]. The application of hydrogels is extensive and encompasses medicine, pharmacy, personal care, agriculture, the food industry, and the environment [3]. Their utilization in the biomedical fields provides novel data about pathological mechanisms or strategies for tissue regeneration and the management of diseases [9,10].

Chitosan (CS) is an amino polysaccharide obtained by deacetylation of naturally occurring chitin from the exoskeleton of shrimps, crabs, and squids [11]. CS demonstrates anti-bacterial and anti-inflammatory characteristics alongside being biocompatible and biodegradable. Its hemostatic nature, ensured by its high charge, is advantageous since it stimulates erythrocyte adhesion, fibrinogen adsorption, and platelet activation [12,13], which paved the way for multiple studies of chitosan-based formulations on wound healing [14,15]. Their biocompatibility, biodegradability, and the ability to encapsulate and release the drug make them attractive and highly applicable molecular carriers. Thus, it is no wonder that chitosan-based formulations are employed in neurological disorders, diabetes, tumors, and cardiovascular diseases [16,17,18,19].

Hydrogels are being constantly improved to provide superior clinical results. Among the upgrades, stimuli-responsive formulations are worth exploring since they react and adjust in response to various external stimuli such as pH, temperature, light, electric field, or ions [18]. Some are developed with multi-responsive properties to react to stimuli such as pH and reducing or oxidizing agents. The triggering stimuli for these hydrogels can occur simultaneously and independently, contingent upon the design and intended application [20]. Their capacity to undergo changes when exposed to stimuli enables highly controlled release patterns. Furthermore, stimuli-responsive hydrogels can mimic intrinsic tissues’ mechanical, biological, and physicochemical characteristics. These unique features position stimuli-responsive hydrogels at the forefront of innovation, facilitating more efficient and safer strategies. Since there is constant progress in the field, it is necessary to provide an up-to-date summary of current trends and breakthroughs in the biomedical application of stimuli-responsive chitosan-based hydrogels, which was the aim of this review. The present paper constitutes a valuable resource for researchers and practitioners.

## 2. Manufacture, Structure, and Administration of Hydrogels

### 2.1. Manufacturing Hydrogels

Hydrogels are manufactured through diverse strategies tailored to the type of intended application, such as 3D printing in tissue engineering. Formulations are divided depending on the number of monomers and their arrangement, which dictates their further application. Manufacture typically involves polymerizing a given compound and cross-linking reactions between the components. This results in diverse types of structures with a wide range of applications (Figure 1) [21,22,23]. Dynamic covalent bonds provide the strong yet dynamic properties of hydrogels. Pectin is used to acquire better control over the release of drugs and to design supramolecular hydrogels with high molecular complexity, detailed as non-contact and contact stimuli-responsive hydrogels [24,25,26,27]. Examples of non-contact stimuli-responsive formulations are photo-sensitive hydrogels whose production utilizes light-responsive compounds, such as azobenzenes, diarylethenes, coumarins, and nitrobenzyl derivatives. Another subset includes thermo-responsive hydrogels that use lower critical solution temperature and upper critical solution temperature to shift between concentration phases [28,29,30,31]; their production utilizes N,N-dimethylacrylamide, ammoniopropane sulfonate, or poly (2-ethoxyethyl vinyl ether) [32,33]. Stimuli-responsive hydrogels are commonly used in medicine; they can be produced using Schiff base cross-links or DNA triplex structures that respond to environmental changes [34,35]. In general, there is an extensive diversity of techniques available for creating hydrogels, spanning from molecular assembly to cutting-edge 3D printing methods. These techniques are customized to suit the particular type of hydrogel being produced and its intended use [25,36].

### 2.2. Structure of Chitosan and the Process of Obtaining Hydrogels

Despite their similar structure, hydrogels can be differentiated based on their origin into synthetic, semi-synthetic, and natural ones. While the choice of the source varies depending on specific applications and desired properties, natural polymer-based hydrogels are often favored over synthetic ones due to their biocompatibility and safety [37]. The human body contains hydrogel structures such as collagen, gelatin, cartilage, mucous, epidermis, meniscus, tendons, and vitreous humor [38]. Biomaterials are widely observed in nature and can be produced using, e.g., various saccharides built of monosaccharide units (homopolysaccharides) such as cellulose, agarose, or chitin. The latter is considered one of the most promising in the development of novel medical solutions [22,37].

Chitin is a natural carbohydrate in the exoskeletons of, e.g., crabs, lobsters, shrimps, and squids. Its repeating units are linked by β-glycosidic linkages. Through its deacetylation in alkaline media, it is possible to obtain chitosan that has functional groups such as C_3_–OH, C_6_–OH, and C_2_–NH_2_, as well as acetylamino and glycoside bonds (Figure 2) [37]. Their modification ensures desirable physical and chemical properties for the specific purpose and type of further administration [39]. Homopolymers of chitosan cross-linked together create a molecular lattice. Its hydrophilic and 3D structure determines permeability for bioactive substances [22], which enables it to retain significant amounts of water and biological fluids. Chitosan homopolymers maintain the integrity of their network and do not dissolve in water at neutral pH, but they can be dissolved in acidic conditions due to the protonation of free amino groups [40].

### 2.3. Routes of Administering Hydrogels

Ongoing efforts provide novel drug delivery approaches to ensure suitable concentrations for therapeutic responses. Some medicines are quickly processed in the body, so they need to be administered often to maintain the appropriate levels in the blood, which is less convenient for patients and poses a risk of missing a dose. Ideally, the therapeutics should provide quick response, work for a long time, and stay at an adequate level. Chitosan and its nanoparticle derivatives assist the process by slowly releasing medicine over time, facilitating usage, and reducing unwanted effects [39]. Utilizing chitosan-based and other kinds of hydrogels for drug delivery has been explored through various administration routes, such as the transdermal route, in which hydrogels form a wound dressing that releases bioactive compounds to recreate natural conditions and promote healing [41]. This approach can also deliver the drug through the skin into the bloodstream. Aided by iontophoresis and electroporation, the transdermal route has enabled increased absorption of compounds such as insulin and vasopressin [42]. Three-dimensional molecular networks can also be administered to the patient through injections, which allow for precise delivery of the substance to the targeted area, such as the knee joint [43]. Another route is oral administration since hydrogels can effectively cope with the low pH of the stomach or even release the active substance in response to the acidic environment [44]. Biomaterials can also be delivered via ocular, nasal, vaginal, and rectal administration [45,46,47,48].

### 2.4. Advantages of Chitosan-Based Hydrogels

Incorporating chitosan in hydrogels ensures improved biocompatibility and biodegradability, as well as elevated resistance to deformation and collapse [49,50]. Chitosan-based hydrogels have anti-bacterial abilities linked to the chitosan’s positively charged amino groups that can interact with negatively charged components of the bacterial cell wall, increasing permeability and leading to cell damage [51]. Furthermore, chitosan has a high charge, which can promote platelet activation, fibrinogen adsorption, and erythrocyte adhesion. It also has a hemostatic effect, which improves wound healing [12,13]. Since hydrogels have properties similar to liquids, molecules may be readily diffused, absorbed, and released [6]. Adding chitosan into the hydrogel structure makes it more hydrophilic, allowing it to swell and expand in the presence of liquid [40,52]. In addition to chitosan concentration, examples of key agents influencing the physicochemical properties of these hydrogels are cross-linkers (e.g., genipine, glutaraldehyde) and hydroxy acids (e.g., mandelic acid, lactobionic acid), which affect morphology, swelling ability, viscosity, mechanical strength, or cytotoxicity [53,54,55,56].

Chitosan was compared to its derivatives or alternative hydrogel-forming compounds. One of the studies utilized chitosan–graft–glycerol and carboxymethyl chitosan–graft–glycerol hydrogels to reduce the side effects of vincristine sulphate; both showed a high swelling ratio and sustained release behavior, indicating their effectiveness in drug delivery to breast cancer cells. Encapsulation efficiency was better in hydrogels based on chitosan (72.28–89.97%) than its derivative (56.97–71.91%) [57]. Another study compared chitosan hydrogel with povidone–iodine hydrogel, revealing similar anti-bacterial activity but a higher in vivo wound closure when using chitosan hydrogel. Additionally, the latter enhanced collagen deposition and decreased the production of proinflammatory cytokines, indicating that the chitosan formulation is an effective alternative to povidone–iodine hydrogel [58].

Chitosan-based hydrogels can also serve as a base for additives or as a standalone compound combined with other gels. Scientists eliminated proteins, pigments, mineral salts, and waxes from the bodies of naturally occurring dead honeybees to produce an exoskeleton-derived chitosan termed Beetosan^®^. Following the hydrogel preparation and introduction of nanoadditives for biocidal activity, researchers measured the swelling, surface characteristics, sorption capacity, cell interaction, and behavior in simulated body fluids such as Ringer’s liquid or artificial saliva. Although chitosan and Beetosan^®^ hydrogels had similar functional characteristics, their structure differed—a certain amount of chitin was retained in the pores of Beetosan^®^, which decreased sorption capacity due to the clogging of pores after introducing silver nanoparticles. Nonetheless, the use of naturally expired honeybees in synthesizing hydrogel materials was noted as a remarkable step due to the possibility of utilizing waste materials [59]. Another study investigated Aloe vera leaf gel and chitosan nanoparticle thin-film membranes to heal wounds infected with methicillin-resistant *Staphylococcus aureus*. Thirty rats were randomized into five treatment groups that received 0.9% saline solution, saline, chitosan nanoparticle thin-film membrane, Aloe vera leaf gel, or Aloe vera leaf gel in combination with chitosan thin-film membranes. Compared to the separate treatments, Aloe vera leaf gel combined with chitosan nanoparticle thin-film membranes decreased the maturation time of granulation tissue, enhanced reepithelialisation, improved collagen deposition, and reduced bacterial count. The authors concluded that Aloe vera leaf gel with chitosan nanoparticle thin-film membranes has reproducible wound healing potential and the potential to be a viable treatment [60].

## 3. Selected Types of Stimuli-Responsive Hydrogels

Hydrogels respond to various stimuli, which allows for dynamic control over their properties and behavior in a wide array of applications. Consecutive subsections summarize formulations responsive to one or more stimuli such as ions, pH, redox potential, light, electric field, temperature, and magnetic field, which are recapitulated in Figure 3.

### 3.1. Ion-Responsive Hydrogels

Formulations sensitive to ions utilize homeostatic ionic strength and composition to trigger drug release, phase transition, or diagnostic signal emission [61]. Furthermore, they may be useful in treating diseases with dysregulated ionic concentrations or with toxic ions present. Elevated calcium ion (Ca^2+^) levels in the bloodstream have been associated with a range of vascular and bone disorders. Zinc ion (Zn^2+^) concentrations are notably higher in nervous tissue, whereas fluctuations of iron ion (Fe^3+^) are related to anemia [61,62,63]. Ion-responsive hydrogels can be implemented to counteract potentially hazardous heavy metal contamination caused by technological advancements. Designing carriers for chelating agents that are released in response to specific toxic ions may be valuable in industrial settings [64]. Regarding water contamination, there has been research on mercury ion detection and its removal from wastewater with the use of hydrogels containing a rhodamine derivative and four different chelating agents [65]. Regarding anion-responsive formulations, a study by Wang et al. highlighted that fluoride (F^−^) and chloride (Cl^−^) prompt the gel to collapse because of their strong hydrogen atom binding, whereas iodide (I^−^) causes a color alteration in the gel without complete degradation [66,67]. Chitosan-based hydrogels exhibit gelation and swelling behavior that can be altered by ionic strength [68,69]. A chitosan hydrogel actuator was created by Zhu et al. using an anodic electrical writing process. The gradient structure of hydrogel allowed it to bend autonomously, with opposite bending in deionized water or NaCl solution [70].

### 3.2. pH-Responsive Hydrogels

Formulations are responsive to pH function because hydrogen ion concentration varies depending on the physiological compartment [71]. Such hydrogels are composed of a primary polymer structure that exhibits weak acidic or basic groups. These groups become more charged in environments with fluctuating pH levels [72]. This affects ionic strength and leads to gel shrinking or swelling. In an alkaline environment, acidic group-containing formulations such as carboxymethyl chitosan hydrogels undergo ionization. Coupled with electrostatic repulsion, this induces a phase transition characterized by drastic swelling [73,74]. Hydrogels containing basic residues operate on an analogous principle, where they release the drug into the target area in an environment with low pH due to fixed negative charges on the polymer and positive charges in the solution [74]. Hydrogen responsiveness is useful in oral drug delivery since the gastrointestinal tract is characterized by a very low pH, which often disrupts drug absorption in the stomach [71]. The pH-sensitive swelling of chitosan hydrogel involves the protonation of its amine groups under low pH conditions [75], which was found useful in drug delivery [76,77]. El-Mahrouk et al. successfully constructed a metronidazole-containing chitosan formulation that had the ability to remain in the stomachs of dogs and to release the drug in a controlled manner. The study demonstrated greater efficacy of hydrogel compared to the classical method of drug administration [78]. Another application was proposed by Ezati et al., who developed a composite film that changed color in response to a pH of 4 to 10. The authors suggested that chitosan-alizarin film can be used for smart food packaging [79].

### 3.3. Redox-Responsive Hydrogels

Changes in the redox potential are one of the most crucial triggers in biomedical research. A balanced redox state is critical in many physiological processes, such as proliferation, differentiation, cellular death, gene expression, mitochondrial activity, Ca^2+^ regulation, immunological response, and neural development. In healthy individuals, redox systems such as glutathione and its disulfide (GSH/GSSG) work continuously to buffer reactive oxygen species (ROS) and maintain a stable environment. Patients with cancer, fibrosis, diabetes, and cardiovascular or neurological diseases have dysregulated redox metabolism [80,81,82]. Redox-responsive hydrogels comprise chemically cross-linked polymer networks created by an intermolecular thiol–disulfide exchange reaction between hyperbranched polyamidoamines [81]. Redox responsiveness is achieved through the inclusion of particular chemical components. One such chemical moiety is the disulfide linker, which may be broken down in the presence of reducing agents such as glutathione. Another one that may also respond to reactive oxygen species, such as hydrogen peroxide (H_2_O_2_), is the selenide group [18,83]. Redox sensitivity is frequently integrated into structures that respond to other factors, e.g., pH or light [84]. With the physiological redox environment thoroughly explored, redox-responsive hydrogels may be used in a variety of biomedical applications. In tissue engineering, they can act as biodegradable scaffolds for organ restoration. Active substances such as growth factors can be placed into hydrogel scaffolds and released in response to redox cues, promoting cellular invasion and proliferation [85]. Redox-responsive hydrogels, particularly those formed by in situ gelling methods, are effective in the topical release of medication. Due to the differences in redox potential between intracellular compartments and the extracellular matrix, premature drug release is prevented, and redox-responsive hydrogels dispense the medication inside cells [81]. Redox-responsive chitosan hydrogels are useful in cancer therapy due to their ROS responsiveness [86,87]. These formulations respond to a reductive or oxidative environment, which is utilized in drug release [88]. Chitosan hydrogel can also undergo electrografting to manufacture a catechol-modified formulation that serves as a redox capacitor, which can be further enriched with nanoparticles to amplify electrochemical signals [89].

### 3.4. Photo-Sensitive Hydrogels

Hydrogels can also respond to light—an easily acquired, effective, non-invasive external stimulus with excellent adaptability and versatility [90]. Such formulations usually comprise a polymeric structure and a photo-reactive component [91]. Chromophore absorbs the light and turns it into a chemical signal through a photoreaction such as isomerization (conversion of one isomer into another isomer by light), cleavage (splitting of chemical bonds caused by light), or dimerization (formation of dimers by a photochemical reaction) [92]. Response to light can also result in water absorption or release, leading to volume changes and further swelling or shrinking [93,94]. To initiate a desired reaction, the type and positioning of photo-sensitive moieties must be carefully considered, similar to the selection of an appropriate light source to meet the specific demands of the application. In general, the emission spectrum and the photo-responsive absorption profile should align in an uncontested region of the electromagnetic range in order to increase light penetration depth and the ratio of photons absorbed to those released [95]. Derivatives of o-nitrobenzyl compounds are by far the most commonly utilized photolabile group that has progressed into hydrogel chemistry [96]. Given their photolysis process and adaptable chemical structure, they can function as photocages or photolabile linkers. Absorption characteristics have been tuned by changing the aromatic structure; the initial structure of o-nitrobenzyl ester undergoes photolysis when exposed to 260 nm light, yielding carboxylic acid and o-nitrosobenzyladehyde [97]. Coumarin derivatives are promising as substitutes for o-nitrobenzyl analogs due to their biocompatibility, rapid splitting rates, redshifted absorbance, and processes induced by two-photon absorption. Several modifications of coumarin derivatives have been applied to change their absorption to a medically valuable range [98]. Other photolabile compounds, such as triphenylmethane, p-hydroxyphenyl, and 8-bromo-7-hydroxylquinoline derivatives, have not yet been proven significant for their use in hydrogels [97]. Photo-sensitive hydrogels are utilized in vitro to mimic in vivo conditions—they act as scaffolds, simulating the dynamic nature of biomechanics in live tissues, because they enable distant and contact-free modulation of growth. It is also possible to utilize them in the dimensional and temporary release of medication since 3D networks may encapsulate a variety of pharmaceuticals, and the gel-to-sol transition induced by light can release the medication [99]. These hydrogels are also valuable in the manufacture of adaptable surfaces or soft smart actuators [97,100]. In addition to biodegradability and biocompatibility, chitosan-based hydrogels are characterized by mechanical strength and elasticity, which may be adjusted by the cross-linking process and the addition of various photo-sensitive components [49]. The latter allows it to absorb contaminants, which is valuable in water purification [101].

### 3.5. Electricity-Responsive Hydrogels

The mechanism of action in these hydrogels is based on ions that migrate once the electric current is applied, leading to a phenomenon known as electroosmotic flow [102]. Once the hydrogel contracts, the particles move toward their intended location, often carrying the opposing charge [93]. The conductive abilities of hydrogels arise from their combination with materials that are able to conduct electricity, such as carbon materials (nanotubes and nanofibers), gold or graphene particles, and various metals [103]. Polymers such as polyaniline, polypyrrole, and poly(3,4-ethylenedioxythiophene)-polystyrene sulfonate (PEDOT:PSS) are useful as well [104]. Electroactive hydrogels are characterized by significant water solubility and flexibility, allowing them to change volume and shape by bending, expanding, or scaling down in response to electric stimuli [93,105]. These formulations are useful as smart skin patches because they can modulate ion channels crucial for proliferation, angiogenesis, and tissue regeneration [20]. They can be utilized as injectable vehicles because of their high biocompatibility and maintenance of drug release time, type, and dose. Such an approach is used in the delivery of benzoic acid, amoxicillin, and curcumin [49,106]. Electricity-responsive formulations are also found in bioelectronic devices such as real-time wound monitors or cardiac sensors. Moreover, they are useful in evaluating food quality and safety [89]. Chitosan-based formulations responsive to electricity are polycationic, which makes them a versatile biomedical material. Electroosmosis enables hydrogel swelling on the cathode side while shrinking on the anode side, thus bending to the latter. Chitosan strands will bend back when the polarity of the electric field is inverted [107,108].

### 3.6. Thermo-Responsive Hydrogels

Hydrogels can alter their physical properties (sol-to-gel transition) in response to temperature changes by adjusting the composition of the polymer network [109]. Phase changing begins when a polymer is subjected to stimuli (temperature)—the micelle structure of the polymer switches into a network structure, creating a scaffold [110]. The mechanism of thermo-responsive hydrogels involves the lower critical solution temperature (LCST) and the upper critical solution temperature (UCST). When LCST is present, polymers typically go through a phase separation when the temperature rises above the LCST but return to a single phase when the temperature drops below the LCST. Conversely, UCST polymers tend to experience a phase separation below the UCST but return to a single phase when the temperature exceeds the UCST [111]. One of the most extensively exploited thermo-sensitive hydrogels is based on poly(N-isopropyl acrylamide) and undergoes a reversible phase transition near body temperature [112]. Formulations responsive to temperature can also be derived from natural polymers such as chitosan, cellulose, gelatin, collagen, alginate, and hyaluronic acid, which are valuable in medical applications due to their biocompatibility and biodegradability [113,114]. One can utilize these hydrogels in drug delivery, tissue engineering, wound healing, or disease diagnosis and treatment [115,116]. For instance, a thermosensitive injectable chitosan hydrogel was developed for the sustained delivery of disulfiram to cancer cells, showing improved cellular uptake compared to disulfiram alone [117].

### 3.7. Magnetic-Responsive Hydrogels

The incorporation of nanoparticles into the hydrogels allows them to respond to the magnetic field [118]. These particles may self-assemble into an organized structure on the hydrogel surface, which was utilized to regulate cell proliferation, neurogenesis, signal transmission, and extracellular matrix regeneration [94]. Magnetic components can take the form of plated oxides or metallic particles that utilize nickel, iron, or cobalt [118]. Fe_3_O_4_ nanoparticles are frequently used in pharmaceutical settings due to their biocompatibility, strong magnetic ability, and relative simplicity of synthesis [119]. Aside from the type of gel networks and nanoparticles used, the physicochemical characteristics and responsiveness of the hydrogels are impacted by the size and distribution of magnetic components, which affect the interaction with polymer networks. A typical way to create magnetic hydrogels is to mix magnetic nanoparticle suspension with hydrogel precursor solution and then gelatinize under specific conditions [120]. These hydrogels are used in drug delivery since they do not require tactile activation and may be controlled remotely in demanding settings. Under an alternating magnetic field, nanoparticles vibrate, and the temperature of the polymer increases, leading to drug release. Furthermore, magnetic-responsive hydrogels are utilized in the chemotherapy of tumors. Some nanoparticles tend to agglomerate when exposed to a magnetic field, resulting in fewer pores and restricted medication delivery [94]. Many studies have validated the use of these hydrogels in tissue engineering due to their modulating effect on various cells under static or pulsed magnetic fields [121,122,123]. Additionally, magnetic hydrogel-based soft robotics have received increasing attention due to their remote control and unlimited tissue penetration depth [120]. Chitosan-based magnetic-responsive hydrogels can be tuned by adjusting the concentration of magnetic nanoparticles and the cross-linking density of the hydrogel [124,125]. Adjusting their viscoelastic properties ensures controlled drug release [126].

### 3.8. Multi-Responsive Hydrogels

In addition to hydrogels responding to a single stimulus at a time, some formulations can react to many stimuli simultaneously and independently [127]. Their preparation is based on cross-linking via covalent and non-covalent methods [128] or combining desired stimuli into existing hydrogel systems [129]. Multiple-responsive hydrogels are superior to single-responsive hydrogels since they provide a more dynamic and controlled response after application, which makes them more adaptable, susceptible to modification, and tailored to specific needs [130]. Regarding drug delivery, enhanced control of environmental behavior renders more accurate release to a target area, which reduces the manifestation of side effects [20]. For instance, vincristine and dexamethasone were delivered using multi-responsive hydrogels controlled by pH, temperature, and enzyme concentration [57]. Scaffolds regulated by pH, temperature, and ionic strength within the tumor microenvironment allowed drug accumulation in cancer cells. Such an approach showed satisfying results in doxorubicin delivery in vitro [131]. Multi-responsive formulations were also used in central nervous system injuries, bleeding, and bone tissue engineering [14,132,133]. Regarding chitosan hydrogels, multi-responsiveness depends on functional groups introduced into their structure [134,135]. Incorporation of these groups is possible via, e.g., protonation, deprotonation, hydrophilic/hydrophobic balance, chemical cross-linking, graft copolymerization, or blending with other polymers, which offer more functionality and tunability than hydrogels reacting to one stimulus [49,106,134,136,137,138].

## 4. Current Trends in the Biomedical Application of Stimuli-Responsive Chitosan-Based Hydrogels

Multifaceted utilization of smart hydrogels with emphasis on chitosan-based formulations deserves a separate chapter because the application spectrum is vast (Figure 4). Previous sections provided general background on manufacture and stimuli responsiveness, but the data presented below are entirely dedicated to chitosan-based smart hydrogels. Consecutive subsections describe the utilization of these formulations in wound healing, neurological impairments, tissue engineering, and drug delivery in various diseases.

### 4.1. Wound Healing

The phenomena that take place in the human body when a wound is formed involve numerous cells and diverse inflammatory mediators. The tissue recovers to its pre-injury state if no difficulties occur during the wound healing. When the process is dysregulated or halted, there is a possibility of infection or chronic injury [139,140]. It is predicted that 1–2% of people in affluent nations will suffer a chronic wound at some point in their lives and that the percentage will rise over time [141]. Hemorrhage, infection, and amputation are only a few of the numerous consequences of chronic wounds [139,141]. Smart hydrogels can be a great addition to existing treatments since they can be triggered by a patient’s body temperature or the skin’s pH [142,143,144].

Chitosan-based hydrogels have shown beneficial effects on wound treatment in many studies [6,145,146,147,148,149]. One study aimed to design novel 3D bioactive and biomimetic smart hydrogels with oxidized hyaluronic acid (oxHA) and oxidized chitosan (oxCS) as cross-linkers. Three active pharmaceutical ingredients—fusidic acid (FA), allantoin, and coenzyme Q10—were embedded in CS-based hydrogels. Except for FA–CS–oxHA, which exhibited some level of toxicity at higher concentrations, hydrogels were non-toxic; FA–CS–oxHA exhibited anti-bacterial effects against *Staphylococcus aureus*. Overall, the study indicated that non-toxic smart hydrogels with oxidized polymers as cross-linkers are useful for treating irregular deep wounds due to their self-healing and adapting properties [150]. 

Another approach utilized pH-responsive chitosan-based hydrogels for wound healing. Normal skin typically has a pH lower than 5, but the pH of the underlying tissue, which is around 7.4, becomes exposed when the skin is injured. Modifying the gelation process allowed hydrogels with adjustable mechanical properties and varying sensitivity to pH levels to be obtained. Such hydrogels were found to degrade under accelerated basic conditions and enzymatic action yet remain stable in neutral buffers. Minimal toxicity was observed for fibroblasts cultured on these hydrogels. These formulations exhibited a reduction in either cell growth or adhesion compared to tissue culture plastic. No apparent cell mortality was noted, suggesting favorable compatibility with the cells. Overall, hydrogels hold promise for targeted wound closure via pH-dependent responses, potentially accelerating the healing process [151].

Pinho et al. have cross-linked chitosan with hydroxypropyl methylcellulose and 2-hydroxypropyl-β-cyclodextrin to provide caffeic acid-loaded hydrogel. The system released its content in a pH-dependent manner, controlled the proliferation of microorganisms on the wound bed, and prevented wound infection [152]. 

Chitosan hydrogels responsive to pH were also loaded with tobramycin to prevent *Pseudomonas aeruginosa* bacterial growth. Effectivity was evaluated in vitro (L929 cell line from mouse subcutaneous areolar and adipose tissue) and in vivo (burn-injured female Kunming mice). On-demand distribution of the antibiotic was enabled via sensing the presence of bacteria by the hydrogel. Self-healing properties and anti-bacterial characteristics rapidly eradicated *Pseudomonas aeruginosa* in the infected areas [153]. 

A different study focused on wound dressings based on chitosan/guar gum/polyvinyl alcohol, with anti-bacterial, degradable, and pH-responsive properties. The hydrogel was combined with different percentages of non-toxic tetra orthosilicate. These pH-sensitive hydrogels were suitable for controlled drug release and showed decreased swelling with increased ionic concentration in various electrolyte solutions. They displayed strong anti-bacterial properties against both Gram-positive and Gram-negative bacterial strains. Within 140 min, the drug release was 98% in phosphate buffered saline media at pH 7.4 [154].

In some other research by Liu et al., a photoactive self-healing carboxymethyl chitosan-based hydrogel was used to influence multiple tissue repair factors in infected wound healing. The biomaterial had good self-healing ability and mechanical property, protecting a wound against additional harm. It also exhibited good tissue adhesiveness and cell affinity, which enabled compact wound adhesion and stopped bleeding. The authors concluded that this smart hydrogel could prevent bacterial infections under visible light, which led to wound healing via elevated collagen synthesis and re-epithelialization [155].

Mai et al. also focused on a photo-responsive chitosan-based formulation with a porphyrin photosensitizer called sinoporphyrin sodium and fibroblast growth factor encapsulated in poly(lactic-co-glycolic acid). Under mild photoirradiation (30 J/cm^2^, 5 min), a concentration of 10 μg/mL of the designed hydrogel demonstrated potent anti-bacterial and anti-biofilm effects that eliminated nearly 99.99% of *Staphylococcus aureus* and multidrug-resistant strains in vitro. In a burn-infection model, it effectively suppressed bacterial growth and enhanced wound healing. The smart hydrogel treatment increased regenerative factors and reduced proinflammatory factors in burn wounds. These findings highlight this hydrogel as a light-triggered anti-bacterial platform for synergistic therapy of burn infections [156].

Nguyen et al. demonstrated that thermo-sensitive hydrogels can also be applied to wound healing. The experiment included the integration of cellulose nanofiber oxidized by 2,2,6,6-tetramethylpiperidine-1-oxyl radical into chitosan and indicated anti-inflammatory and wound healing properties at 14 days post-implantation [157]. 

Ma et al. evaluated the photothermal activity of an injectable thermosensitive composite of hydroxypropyl chitin, tannic acid, and ferric ions (Fe^3+^). The hydrogel showed good cytocompatibility and hemocompatibility even with a low dose of tannic acid, an anti-bacterial agent. Both in vitro and in vivo tests have demonstrated bactericidal effects within 10 min of near-infrared laser exposure. Combining low-level laser therapy with hydrogel promoted tissue repair, suggesting significant potential for clinical application in infected wounds [158].

Multi-responsive chitosan-based hydrogels were also investigated in wound healing. One in vitro study assessed hydrogels responsive to temperature and pH, which were loaded with gentamycin sulphate and used as dressings. A substantial amount of chitosan contributed to pH responsiveness, while including poly(N-isopropylacrylamide) enhanced temperature sensitivity. The film inhibited the Gram-negative bacterium *Escherichia coli* under a high pH and temperature [159].

### 4.2. Neurological Impairments

Neurological impairments are a considerable burden on an aging society. They are caused by brain trauma, neural degeneration, untreated bacterial/viral infections, and chronic inflammation [160]. Although current therapeutic approaches do not yield satisfactory outcomes, it is essential to note that administering medications and finding novel remedies for neurologic disorders are complicated by the blood–brain barrier, limited accessibility to brain tissue, and high levels of patient discomfort during therapeutic operations [161,162]. 

Owing to their biocompatible, biodegradable, and non-toxic features, smart chitosan hydrogels were implemented in several studies on neurological impairments. The thermo-responsive hydrogel was employed as an ibuprofen delivery carrier administered via the nasal route to ensure high efficiency of drug release into brain tissue. Thermal response was obtained by combining chitosan with β-glycerophosphate disodium salt. The prepared solution was transformed into a spray using a VP3 Aptar pump, and dispersion efficiency was assessed. Biological features of the chitosan–ibuprofen hydrogel were assessed using nasal epithelial CCL-30 cell culture. High intranasal bioavailability and no cytotoxicity for nasal cells were noted [163]. Administration via the nasal route was also used to deliver ropinirole, a dopamine receptor agonist [164]. Khan et al. analyzed the intranasal solution activity on albino Wistar rats and obtained similar results as described above [165]. 

Self-healing chitosan hydrogels were also implemented to trigger central nervous system repair on a stroke model. Liu and colleagues presented chitosan–hyaluronan hydrogel responsive to pH, temperature, and light. Cytotoxicity was assessed in vitro using neural stem cells harvested from adult mouse brains. Cell proliferation and viability were evaluated to determine if hydrogel ensures a favorable environment for neural regeneration. The functioning of chitosan–hyaluronan hydrogel was also assessed on two in vivo models, i.e., zebrafish traumatic brain injury and rat intracerebral hemorrhage. Hydrogel was injected into the skull with a needle under tricaine anesthesia or into the striatum under isoflurane gaseous anesthesia, respectively. Motor function recovery and central nervous system repair were established by the swimming-forward test for zebrafish. The brains of rat models were examined with magnetic resonance imaging, histological analysis, and real-time PCR. The developed hydrogel was found to positively influence the renewal of the central nervous system, which holds promise for those who do not benefit from conventional therapies [166].

Nanoformulations incorporating chitosan are increasingly being used to improve treatment. Similarly to chitosan hydrogels, nanoparticles can react to different stimuli and are characterized by biocompatibility and biodegradability. In a study on Alzheimer’s disease (aluminium chloride-induced rat model), Almuhayawi et al. used nanoformulation to encapsulate pomegranate extract that exhibits neuroregulatory and anti-oxidant properties. Nanoparticles were prepared through magnetic mixing of chitosan, ethanol, lecithin, and pomegranate extract, then lyophilized. The effectiveness of pomegranate extract was evaluated by the novel object recognition test, whereas molecular analysis was carried out on isolated rat brains. Data imply that using chitosan as a nanocarrier for pomegranate enhanced levels of anti-oxidant biomarkers (catalase and reduced glutathione) and total anti-oxidant capacity, improving the effectiveness of an active compound [167]. Nanoformulations incorporating chitosan were also effectively used to supply risperidone in schizophrenia, as well as to deliver niruriflavone, tacrine, and resveratrol in Alzheimer’s disease [168,169,170,171,172].

### 4.3. Tissue Engineering

Tissue engineering is a revolutionary, fast-developing field addressing disabilities that stem from tissue loss or damage. Converting biological, chemical, and physical knowledge into valuable materials, devices, and therapeutic approaches allows for replacing, repairing, or regenerating organs and tissues [38,173,174,175]. Tissue engineering employs cells, scaffolds, growth-stimulating signals, and bioreactors [176]. Hydrogels are usually but not always enriched with cells, and their injection into the defective site stimulates regeneration [177]. Consecutive paragraphs summarize the utilization of chitosan-based smart hydrogels in engineering bone and neural tissue, respectively.

Some studies focus on implementing smart chitosan hydrogels for regenerating cartilage and bone [178,179]. The injectable hydrogel made from chitosan and dextran aldehyde showed adaptability in manufacturing and mechanical properties. The release of bovine serum albumin and mechanical strength varied with cross-linking extent. Encapsulated osteoblasts maintained high viability and growth [180]. Different studies used a drug delivery system where kartogenin was released from poly(lactic-co-glycolic acid) microspheres in a chitosan–chondroitin sulfate hydrogel upon ultrasound exposure. This enabled non-invasive delivery to enhance chondrogenic stem cell differentiation for cartilage tissue engineering. Drug release was slow, lasting over 28 days. Cell viability was not affected, and kartogenin-loaded microspheres increased collagen synthesis. Rheological analysis confirmed the structural integrity of a scaffold [181]. Another experiment applied chitosan–hyaluronic acid with silanized hydroxypropyl methylcellulose as an injectable hydrogel for cartilage tissue engineering. The study demonstrated that the formulation containing 3% silanized hydroxypropyl methylcellulose displayed the most favorable characteristics suitable for regenerative applications [182]. Huang et al. created a composite scaffold that consisted of magnesium oxide nanoparticle-coated eggshell particles and chitosan. It was used to deliver magnesium ions and bone morphogenic protein-2 to promote osteogenesis in vitro and in a rat calvarial bone defect model. Results indicated significant bone repair with complete filling of bone defects and increased bone density, confirmed by micro-computed tomography and fluorescence quantitative analysis [183]. Temperature- and pH-sensitive chitosan-based hydrogels have also been applied in bone tissue engineering [184]. As an example, Zhao et al. developed a biocompatible system combining carboxymethyl chitosan and amorphous calcium phosphate with glucono-δ-lactone, which effectively increased the expression of bone markers such as runt-related transcription factor-2, osterix, osteocalcin, and osteopontin. Moreover, it supported proliferation, adhesion, and osteoinduction. Significant improvements in bone regeneration efficiency and inhibited bone resorption were noted [185]. Niranjan et al. manufactured and analyzed thermo-sensitive hydrogel, which consisted of zinc, chitosan, and β-glycerophosphate, for bone engineering purposes. The hydrogel showed swelling ability, anti-bacterial activity, and promotion of osteoblast differentiation [186]. Another study utilized an injectable composite containing modified halloysite nanotubes that improved the mechanical properties of the chitosan hydrogel and facilitated the differentiation of encapsulated bone tissue due to their stiffness, tubular structure, and capability to deliver an osteogenic inducer agent dynamically [187].

Interestingly, pH-responsive hydrogels used initially in bone tissue engineering were found useful in neural tissue engineering [184]. One study utilized injectable hydrogels for remodeling the extracellular matrix [188]. Not only thermo-responsive and pH-sensitive formulations can be used in neural tissue engineering [189,190,191], but also those responding to the electric field are beneficial because they rapidly deform when subjected to conditions such as ion currents from streaming potentials, mimicking the natural environment [38]. CoFe_2_O_4_, Fe_3_O_4_, or Fe_2_O_3_ are typically incorporated in magnetic-responsive hydrogels to provide rapid response, biocompatibility, and desirable mechanical properties [192,193,194]. Other hydrogels, such as ion-responsive or ultrasound-sensitive, are also utilized in neural tissue engineering [193,195,196]. Liu and colleagues used chitosan scaffolds loaded with the nerve growth factor to address sciatic nerve defects in adult rats. Not only was the functional formation of neural circuits achieved, but the reconnection of nerves to target muscles also restored contractile function [197]. There are assumptions that chitosan can promote nerve regeneration by enhancing the proliferation and migration of Schwann cells [198,199]. Guo et al. constructed hollow conduits and infused them with a hydrogel comprising simvastatin and Pluronic^®^ F-127 to assess their efficacy in peripheral nerve regeneration. The damaged sciatic nerves were regenerated using chitosan conduits with varying concentrations of simvastatin [200]. A different study evaluated chitosan conduits enhanced with fibrin–collagen hydrogel and adipose-derived mesenchymal stem cells for repairing sciatic nerve defects. Sensory and motor recovery, muscle atrophy, gene expression, and histological changes were assessed. Sensory recovery was observed in all experimental groups, and upregulation of regenerative genes was noted, suggesting potential for improved nerve regeneration. However, motor recovery was limited, and immunohistochemistry showed varying levels of nerve regeneration [201]. Another chitosan-based hydrogel was enriched with mesenchymal stem cells to promote spinal cord regeneration by limiting glial scar formation and decreasing cell death at the injured site. Hydrogel supported the paracrine activity of mesenchymal stem cells in treating spinal cord injury, which led to improved functional recovery in a mouse model. Good biocompatibility, injectability, and permeability were demonstrated. The highlighted challenges included the poor survival rate of transplanted mesenchymal stem cells within the harsh environment and unregulated cell migration [202]. Lastly, Stanzione et al. developed a thermosensitive chitosan-based hydrogel combined with β-glycerophosphate and sodium hydrogen carbonate, which allowed optimal mechanical properties and injectability. All formulations maintained the viability of the hybrid cell line NSC-34 containing neuroblastoma and spinal cord cells, which allowed for cell differentiation characterized by prolonged axotomy. These findings suggest that hydrogels are a promising strategy for disorders affecting motor neurons [203].

### 4.4. Drug Delivery in Cancer Treatment

Drug delivery comprises the strategy and formulation of the medication, manufacturing and storage techniques, and administering methods. Although it is evolving quickly and efficiently, some approaches fail to achieve the desired treatment outcomes, causing toxicity and putting safety at risk. A formidable obstacle to providing complete cancer remission is the complicated nature of the disease, which emphasizes the need to improve the survival and recovery rates of oncological patients [204,205]. Stimuli-responsive hydrogels may act as drug carriers, e.g., in the form of injectable compounds [206].

An interesting approach was applied by Khan and colleagues, who designed a thermo- and pH-responsive hydrogel made of carboxymethyl chitosan and poly(N-Isopropylacrylamide). The formulation was loaded with 5-fluorouracil, which is commonly used in the chemotherapy of esophageal, gastric, colorectal, breast, and cervical tumors [207,208]. In vitro models were investigated using MTT assay, half maximal inhibitory concentration (IC50), and cytocompatibility assay. Chitosan-based hydrogel was found useful in localized drug delivery and controlled drug release after subcutaneous administration [207]. 

Smart hydrogels were also utilized to deliver doxorubicin, which is widely used in lymphoblastic/myeloblastic leukemia, as well as lung, breast, bladder, and colorectal cancer [209]. Qian et al. combined carboxymethyl chitosan with doxorubicin-loaded zeolitic imidazolate framework nanoparticles for intra-tumoral injections, presenting non-toxic trades and considerable drug release in an acidic tumor microenvironment [210]. Another team enriched chitosan with polyethylene glycol to obtain a pH fluctuation response; doxorubicin was encapsulated in the hydrogel with the addition of sodium bicarbonate, which allowed for a shorter gelation time. This non-toxic formulation neutralized local acidosis and presented efficient chemotherapeutic release [211]. Similar effects for doxorubicin delivery were obtained by Qu et al. in primary liver cancer using N-carboxyethyl chitosan pH-sensitive hydrogel [212], Wang et al. in melanoma by incorporating chitosan–MnO_2_ hydrogels responsive to redox and light [213], and Meng et al. in prostate cancer with O-carboxymethyl chitosan/perfluorohexane nanodroplets reactive to pH and ultrasound [214].

Stimuli-responsive hydrogels can also improve administering another anti-cancer drug, i.e., disulfiram [215]. Ahsan and colleagues proposed an injectable, thermo-responsive chitosan hydrogel encapsulating system. The efficiency of drug release was checked on the dialysis membrane model in vitro; administering disulfiram via chitosan hydrogel improved anti-cancer treatment [117]. Bazzazzadeh et al. used chitosan-based carriers in temozolomide and paclitaxel delivery against glioblastoma. Chitosan hydrogel was combined with poly(acrylic acid), polyurethane, and an aluminum terephthalate-based metal organic framework to create nanofibers. Anti-cancer drugs were loaded into the nanofiber using the electrospinning method. The composite was sensitive to magnetic field, pH, and temperature to enable more precise and effective drug delivery. The amount of glioblastoma cell line U87-MG was greatly reduced, and the drug was characterized by increased stimulation of cell death. This kind of strategy may overcome issues with blood–brain barrier bypassing because the medications are delivered locally [216].

Chitosan hydrogels have been used to administer vincristine, which has also decreased its adverse effects. A study implemented chitosan–graft–glycerol hydrogel and carboxymethyl chitosan–graft–glycerol hydrogel, which were synthesized by combining all compounds with a magnetic stirrer. Hydrogels were found safe, non-toxic, biocompatible, and pro-apoptotic using IC50, MTT assay, Annexin-V assay, and flow cytometry in the breast cancer cell line MCF-7. The study indicated promising encapsulation properties, local delivery capacities, and efficient drug release [57].

Chitosan formulations have been used to transport natural and herbal components, of which compounds such as curcumin and quercetin are useful in counteracting cancer [217,218]. Li et al. studied the dual-stimuli chitosan hydrogel for curcumin distribution in solid tumors. Curcumin was encapsulated into liposomes and introduced to a thermo- and pH-sensitive thiol-derivatized chitosan hydrogel. Carriers were added to the culture of MCF-7 cells in order to monitor growth and cytotoxicity. Decreased cell viability was noted alongside satisfactory drug release in the acidic pH that imitated the tumor microenvironment [219]. Baghbani and colleagues showed promising utilization of chitosan/perfluorohexane polymer for smart curcumin delivery to breast cancer cell line 4T1. The study indicated efficient ultrasound-induced drug release in dialysis tubing, anti-tumor activity by MTT assay, and non-toxicity through hemolytic tests [220]. Regarding quercetin, Sabourian et al. manufactured chitosan–hyaluronic acid nanoparticles responsive to pH and oxidative stress. This approach confirmed cell death in glioblastoma culture, as well as the high capacity of drug delivery and release [218].

### 4.5. Drug Delivery in the Treatment of Non-Cancer Diseases

It is worth recapitulating that chitosan formulations may be used for drug delivery in diseases other than cancer. Smart hydrogels are valuable in treating gastric ulcers, which are sores that develop on the lining of the stomach and are characterized by burning or gnawing pain in the center of the abdomen. The chitosan-based pH-responsive hydrogel was used for the controlled release of amoxicillin, a drug used in gastric ulcer treatment. Hydrogel was found to enhance mechanical properties owing to the encapsulation of bone ash into a hydrogel, ensuring its integrity until it reached the targeted drug-release region. No significant cytotoxic effects on cell viability were noted [221]. 

Regarding gastric ulcers, one of the contributing factors worth mentioning is *Helicobacter pylori*. Lin et al. have designed pH-responsive chitosan–heparin nanoparticles against *Helicobacter pylori*. The formulation was stable at pH 1.2–1.5, which allowed for avoiding the adverse effects of gastric acids and providing a drug-protective environment. Nanoparticles infiltrated the cell–cell junctions and interacted with infected sites in the intercellular spaces [222].

Another application of hydrogels is related to the degenerative diseases of bones or joints. Naproxen is a drug used in many conditions, such as acute gout, ankylosing spondylitis, bursitis, polyarticular juvenile idiopathic arthritis, osteoarthritis, tendonitis, rheumatoid arthritis, and primary dysmenorrheal [223]. Jiang et al. synthesized bacterial cellulose–chitosan zwitterionic hydrogels with pH responsiveness for the drug release of naproxen. This formulation showed minimal swelling at pH 3.5–5.0 and increased swelling at lower or higher pH levels. Hydrogel was able to load naproxen at a rate of over 110 mg/g, with sustained release for more than 24 h and preferential release in intestinal fluid [224]. 

One of the therapeutic options for rheumatoid arthritis is phenethyl isothiocyanate. Despite its anti-oxidant and anti-inflammatory features, it may be characterized by low water solubility, short half-life, and instability, resulting in low biocompatibility [225,226]. Haloi et al. loaded phenethyl isothiocyanate into thermo-sensitive chitosan–Pluronic F-127 hydrogel and analyzed its efficiency as a delivery system in chronic inflammatory conditions. It demonstrated strong mechanical strength, injectability, biocompatibility, and biosafety for sustained drug release. Local administration of this hydrogel significantly reduced inflammation markers in a rat model of arthritis, overcoming the limitations of phenethyl isothiocyanate, i.e., thermal instability and rapid clearance. Incorporating phenethyl isothiocyanate into the hydrogel allowed for prolonged therapeutic activity at sub-toxic levels, highlighting the potential of polysaccharide-based hydrogels as carriers for various phytochemicals with similar limitations [227].

Glaucoma is a neuropathy characterized by loss of the visual field and blindness caused by the destruction of ganglion cells [228,229]. While clinical trials have shown promising results in reducing intraocular pressure, existing non-hydrogel methods entail adverse effects such as ocular inflammation, itching, or visual disturbances from topical treatments [228,230,231]. Smart chitosan-based formulations have emerged as a potential solution for glaucoma since they are known for their biocompatibility and ability to absorb water [232]. Some applications focus on hydrogels as drug carriers combined with other polymers, such as chitosan or gelatin, while others evaluate stimuli-responsive hydrogels [233].

Another experiment aimed at creating a non-invasive alternative to eye drops for glaucoma treatment. Cheng et al. used thermo-sensitive chitosan-based hydrogel for topical application. Results indicated biocompatibility and sustained-release profiles in vitro and in vivo. After weekly application of hydrogel, the intraocular pressure significantly decreased within seven days and maintained a consistent level throughout the duration of the study [234]. 

Pakzad et al. also focused on a thermo-responsive chitosan-based hydrogel as a topical ocular delivery system. The formulation was loaded with timolol maleate as a drug for glaucoma. The study confirmed the biocompatibility and degradability of hydrogel, which released the drug in a constant linear manner for over a week [235]. 

Another biomedical application is related to biosensors, which are electronic devices capable of monitoring reactions, treatment, and disease progression. Biosensors have the structural ability to include nanocomponents. This kind of equipment provides better signaling capacities, reduced production costs, and improved patient comfort [236]. One of the studies described an on-skin alcohol biosensing system to monitor alcohol levels from sweat. The device was made of two parts, i.e., the iontophoretic electrodes responsible for inducing sweat by delivering the pilocarpine drug and three amperometry electrodes (working, reference, and counter) located inside the anode. Chitosan was a part of the reagent layer involved in the amperometric sensing of ethanol on the working electrode. Detection of ethanol by the Prussian Blue transducer triggered the flexible electronic circuit, which sent data about the alcohol intake via Bluetooth. This novel approach holds promise for non-invasive monitoring of individuals [237]. 

It is possible to include chitosan as a component of the wearable strain sensor for capturing human motions and physiological signs in real-time applications utilizing artificial intelligence, soft robotics, and biomimetic prostheses. Xia et al. developed a polyacrylamide–chitosan hybrid network where polyacrylamide was cross-linked by hydrophobic association, while the chitosan network was formed by ionically cross-linking carboxyl-functionalized multi-walled carbon nanotubes. The composite showed promising results in the mechanical tensile test, conductive test, peeling adhesion strength, and resistant assay. When the sensor was attached to the body, it was possible to monitor real-time human motions such as speaking, breathing, body movements, and pulse. Hydrogel wearable strain sensors can be a milestone for sensing a broad range of changes among patients [238,239].

Due to their anti-fungal, anti-microbial, biocompatible, and biodegradable characteristics, chitosan hydrogels can be used to treat different types of dermal infections and disorders [240]. Onychomycosis is a highly transmittable and chronically recurrent nail infection that can be caused by several different microbes such as *Candida*, *Trichophyton*, *Microsporum*, and *Scopulariopsis*. Bozoğlan and colleagues used chitosan, carboxymethylcellulose, scleroglucan, and montmorillonite to create hydrogel nanocomposite. It was found that the gelling temperature must be lower than the temperature of the nails to obtain the desired results. The drug release depended on the amount of montmorillonite within the hydrogel matrix and was controlled by changing the amount of clay mineral in the hydrogel material. Promising data were presented regarding oxiconazole nitrate release and anti-fungal properties against *Trichophyton mentagrophytes* and *Trichophyton rubrum* [241].

Conditions such as acute skin injuries and soft tissue infections caused by *Staphylococcus aureus* and *Staphylococcus epidermidis* were also treated with chitosan hydrogels. Hemmingsen et al. utilized two chlorhexidine-containing chitosan-based systems, i.e., chitosan-infused vesicles alone or the same vesicles incorporated in a hydrogel. Cytotoxicity and biocompatibility were investigated using HaCaT cells, which represent human keratinocytes. The combination of chitosan and chlorhexidine resulted in an anti-bacterial effect that was quicker and lasted longer [242]. 

Frade et al. evaluated the efficacy of the anti-microbial activity of chitosan-based hydrogel alone and in combination with the methylene blue associated with photodynamic therapy against the planktonic and biofilm phase of *Propionibacterium acnes* that cause acne vulgaris. Formulations were adhesive to the skin, and including methylene blue did not influence the bioadhesive force. The synergistic effect of chitosan and photodynamic therapy was observed only in the planktonic phase tests. The authors suggested the utility of a topical chitosan-based system for methylene blue delivery in treating dermatologic conditions with infectious and inflammatory components [243]. A chronic inflammatory state of the mouth mucous layer known as oral lichen planus is currently being investigated in a clinical trial that utilizes erythropoietin-loaded chitosan hydrogels (ClinicalTrials ID: NCT06135259).

Myocardial damage can have various medical backgrounds but is frequently caused by chronic stress, poor diet, smoking, alcohol intake, and comorbidities such as diabetes, hypertension, or hyperlipidemia [244]. Tohidi et al. showed the possible use of a thermosensitive and electroconductive chitosan formulation for regenerating myocardium. Hydrogel was constructed by mixing and grafting chitosan, carboxylated pluronic, and gold nanoparticles. The H9C2 cell culture model (myocardial infarction cells isolated from rat neonates) was used for cytotoxicity and adhesion evaluation in vitro. Biocompatibility, degradability, and stimuli responsiveness confirmed that the nanocomposite is effective in heart tissue repair [245]. 

## 5. Concluding Remarks

The present paper reviewed various stimuli-responsive chitosan-based hydrogels and their biomedical applications, where tremendous progress has been made in the last few years. Although smart chitosan-based hydrogels are already characterized by prominent biocompatibility, biodegradability, and non-immunogenicity [12,13], they are constantly being improved to obtain better results in clinical settings. Stimuli-responsive hydrogels can be produced using Schiff base cross-links or DNA triplex structures; manufacturing hydrogels differs based on their future applications [34,35]. Responsiveness to stimuli may be contact-dependent or independent [24,25,26,27]. Hydrogels are also differentiated based on their origin into synthetic, semi-synthetic, and natural. The latter are often favored over synthetic ones due to their biocompatibility and safety [37].

Hydrogels respond to diverse types of stimuli. In this review, we summarized responsiveness to ions, pH, redox potential, light, electric field, temperature, and magnetic field, as well as hydrogels that are sensitive to two or more stimuli. Ion-responsive hydrogels utilize ionic strength and composition to trigger drug release, phase transition, or diagnostic signal emission [61]. Regarding pH responsiveness, hydrogen ions regulate gel shrinkage or swelling, which is useful in drug administration to a gastrointestinal system in which low pH can impede absorption [71,73,74]. Redox-responsive hydrogels allow drug administration inside cells while preventing premature release; they can be used in tissue engineering and repair, as well as topical medication delivery [81,85]. Photo-sensitive hydrogels respond to light stimuli, creating a polymeric structure useful for simulating biomechanics in living tissues and regulating medicine delivery [91,97,100]. The mechanism of action in electroactive hydrogels is based on ions that migrate once the electric current is applied. These formulations are utilized in dressings, injectable medication administration, and bioelectronic devices [20,89,93,106]. Thermo-responsive hydrogels modify their physical characteristics in reaction to temperature changes, which helps in tissue engineering, wound healing, and disease detection or therapy [109,115]. Magnetic-responsive hydrogels are made via the incorporation of nanoparticles, facilitating drug delivery, tissue engineering, and soft robotics. These hydrogels can influence neurogenesis, signal transmission, and the extracellular matrix [94,118,120]. Chitosan-based hydrogels may also respond to several stimuli concurrently and independently, resulting in a more dynamic and sophisticated response, as well as negative effect limitation. These formulations are employed in tissue engineering, wound healing, and cancer treatment [14,127,130,131,132,133].

The positive effects of chitosan-based formulations in wound healing have been confirmed by numerous studies [6,145,146,147,149]. Because the current therapeutic approaches for neurological diseases sometimes yield unsatisfactory outcomes, smart chitosan hydrogels are investigated for their potential to solve problems such as limited accessibility or high levels of risk and discomfort during therapeutic procedures [161,162]. Some studies focused on the implementation of smart chitosan hydrogels in tissue engineering related to skeletal and nervous systems [178,179,184,188]. Hydrogels are also known for their drug delivery capacities, which has been utilized in various diseases such as cancer [214], gastric ulcer [221], rheumatoid arthritis [223] or glaucoma [234].

Conclusively, chitosan-based stimuli-responsive hydrogels are advanced materials that exhibit pleiotropic functions in a vast biomedical spectrum, which ensures promising outcomes in the treatment of various conditions. Despite the existence of a considerable amount of research on smart chitosan hydrogels, available data strongly support the idea that it is worth improving current approaches and developing novel solutions.

## Figures and Tables

**Figure 1 gels-10-00295-f001:**
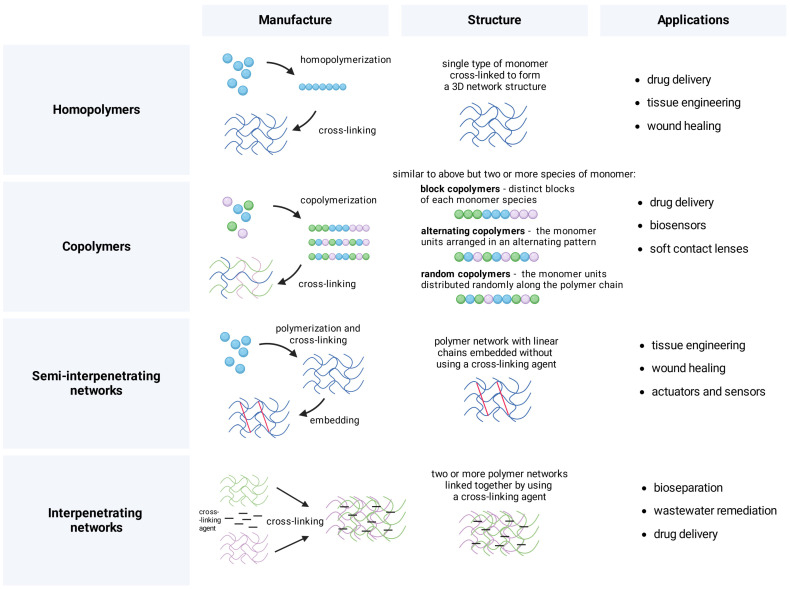
Manufacture, structure, and example applications of polymeric compositions. Formulations are divided depending on the number of monomers and their arrangement, which dictates their further application, e.g., in drug delivery, tissue engineering, and wastewater remediation. Figure created with BioRender.com.

**Figure 2 gels-10-00295-f002:**
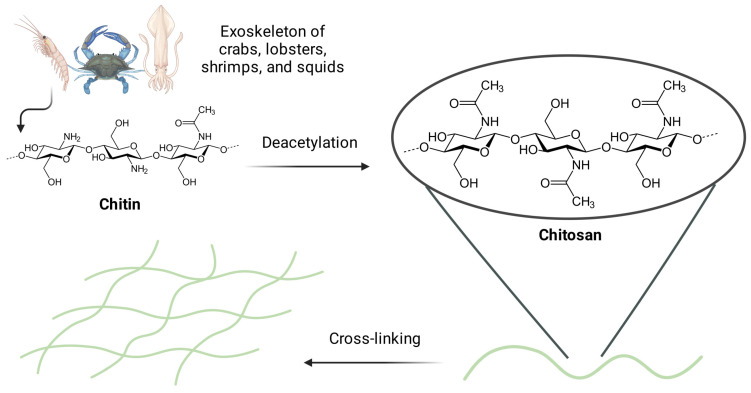
The process of obtaining chitosan-based hydrogels via cross-linking. The exoskeleton of, e.g., crabs, lobsters, shrimps, or squids is the source of chitin that may be deacetylated in alkaline media to obtain chitosan. The latter has functional groups such as C_3_–OH, C_6_–OH, C_2_–NH_2_, as well as acetylamino and glycoside bonds. Figure created with BioRender.com.

**Figure 3 gels-10-00295-f003:**
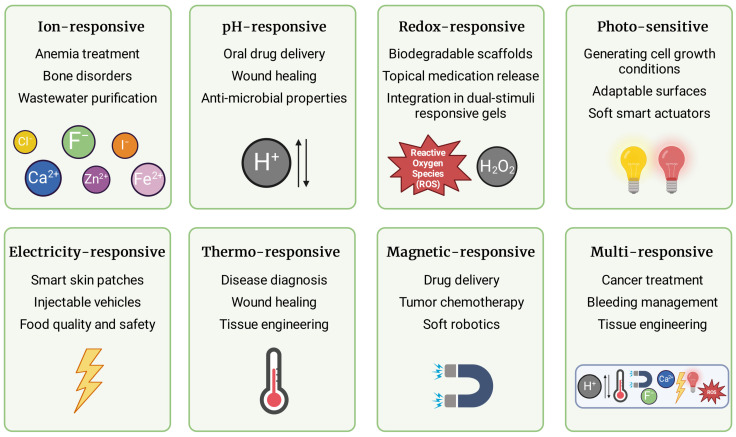
Types of stimuli-responsive hydrogels and their potential application. Hydrogels can react to ions, pH, redox potential, light, electric field, temperature, and magnetic field. They may also respond to several stimuli concurrently and independently. The penultimate chapter of the present paper provides a comprehensive summary of their application. Figure created with BioRender.com.

**Figure 4 gels-10-00295-f004:**
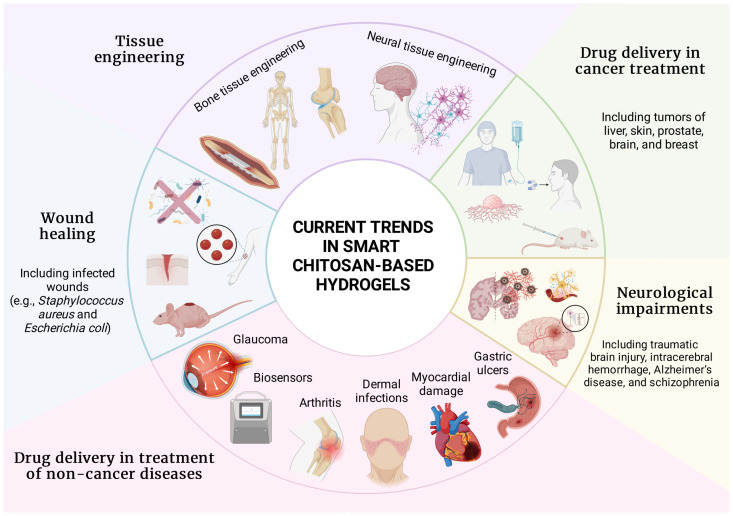
Current trends in chitosan-based smart hydrogels in the field of biomedicine. Chitosan smart hydrogels are emerging in tissue engineering, cancer treatment, wound healing, neurological impairments, and drug delivery in non-cancer diseases, with examples provided for each field. Figure created with BioRender.com.

## Data Availability

No new data were created or analyzed in this study. Data sharing is not applicable to this article.

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
