# Peer review of "Biomedical Trends in Stimuli-Responsive Hydrogels with Emphasis on Chitosan-Based Formulations"

_gels, 2024, doi:10.3390/gels10050295_

Round 1
Reviewer 1 Report
Comments and Suggestions for Authors
The manuscript, “Biomedical Trends in Stimuli-Responsive Hydrogels with Emphasis on Chitosan-Based Formulations”, aims to provide trends in the biomedical application of stimuli-responsive chitosan-based hydrogels.
The review covers various stimuli that can activate hydrogels, including ions, pH, redox potential, photo, and temperature. Additionally, hydrogels that respond to multiple stimuli are mentioned. The review focuses on chitosan-based smart hydrogels and their versatile applications. The authors have provided a well-informed discussion, and I suggest accepting the paper with some minor revisions. Here are my recommendations:
1- However, the article first examines the types of hydrogels, although it is better to clarify, as explained in the following section, about the chitosan-based hydrogels.
2- the text does not match ref 87 in the case of thermo-responsible polymers in lines 287-290. The relevant reference states that these natural polymers can be found in thermo-responsible polymers if they are grafted or cross-linked with temperature-sensitive polymers. However, it was mentioned that “Formulations responsive to temperature can also be derived from natural polymers such as chitosan, cellulose, gelatin, collagen, alginate, and hyaluronic acid, which are valuable in medical applications due to their biocompatibility and biodegradability.”
3- In lines 307-309, the magnetic field is described as a parameter, and its impact is highlighted. However, when the magnetic field rises, the temperature of the thermo-sensitive polymer increases, and the drug is released from the thermo-sensitive hydrogel.
4- Another thing that would be better to add to this article is comparing the effect of adding chitosan to hydrogels with the effect of hydrogels without additives or other additives.
Author Response
Dear Reviewer 1, our response letter has been uploaded as an attachment. Thank you so much for your peer-review activity. Kind regards, Damian Kołat.

Reviewer 2 Report
Comments and Suggestions for Authors
I appreciated the choice of a highly actual topic, smart hydrogels, and in particular chitosan-based smart hydrogels, in which the central place is occupied by chitosan, a biopolymer that has demonstrated over time indisputable qualities for the biomedical field (biocompatibility, biodegradability and non-toxicity), as well as its own pharmacological action - antibacterial, anti-inflammatory, hemostatic, which makes it all the more valuable. The review explores the potential of chitosan and its derivatives in the controlled release of certain drugs, being excellent transport vectors with an important role in improving patient compliance and adherence to treatment.
The review is very well systematized and documented, based on a very extensive number of bibliographical references, most of which are of very recent date, synthesizing information of major interest that are illustrated by representative figures for the chosen subject and the article will certainly prove to be a useful material for researchers who have concerns on this very interesting research field.
I noticed the authors' concerns for this topic based on previous publications, hence the idea and probably the need to update and systemize the information on this research issue so far.
I find this article useful and well received and, as a result, I recommend it for publication.
I congratulate the authors and wish them much success in their future research.
Author Response
Dear Reviewer 2, our response letter has been uploaded as an attachment. Thank you so much for your peer-review activity. Kind regards, Damian Kołat.

Reviewer 3 Report
Comments and Suggestions for Authors
The paper entitled “Biomedical Trends in Stimuli-Responsive Hydrogels with Emphasis on Chitosan-Based Formulations” presents a review topic of real interest regarding Chitosan-based smart hydrogel as biomedical materials.
The paper is informative, with valuable results, however, the authors should consider the following revisions:
- The title of Chapter 2 shows that it will be discussed about "Manufacture, structure, and administration of stimuli-responsive hydrogel", but the authors only exemplify some types of "stimuli-responsive hydrogels", without describing types of obtaining methods, which lead to certain structures of stimuli-responsive formulations. The authors write more in a general way, but they must either describe several examples in the text, or present representative examples in a table, in which the method of obtaining, the realized structure and the application should be emphasized. So, the authors should reorganize Chapter 2.
- The titles of Figure 2 and Figure 3 must be written in a short form, and all other explanations must be placed in the text.
- It must be explained in subsection 3.3. what does the term "ROS" mean in Figure 2.
Although there is a lot of up-to-date work on this topic, to which the authors refer, in order to be a work worthy of publication, the authors should consider reorganizing the chapters.
I believe that the manuscript could be accepted for publication after a minor revision.
Author Response
Dear Reviewer 3, our response letter has been uploaded as an attachment. Thank you so much for your peer-review activity. Kind regards, Damian Kołat.
